# Genome-Wide Identification and Characterization of *HSP70* Gene Family in *Aquilaria sinensis* (Lour.) Gilg

**DOI:** 10.3390/genes13010008

**Published:** 2021-12-21

**Authors:** Cuicui Yu, Mei Rong, Yang Liu, Peiwen Sun, Yanhong Xu, Jianhe Wei

**Affiliations:** 1Key Laboratory of Bioactive Substances and Resources Utilization of Chinese Herbal Medicine, Ministry of Education and National Engineering Laboratory for Breeding of Endangered Medicinal Materials, Institute of Medicinal Plant Development, Chinese Academy of Medical Sciences and Peking Union Medical College, Beijing 100193, China; yucuicui@implad.ac.cn (C.Y.); rongmei@implad.ac.cn (M.R.); liuyang@implad.ac.cn (Y.L.); pwsun@implad.ac.cn (P.S.); 2Hainan Provincial Key Laboratory of Resources Conservation and Development of Southern Medicine, Key Laboratory of State Administration of Traditional Chinese Medicine for Agarwood Sustainable Utilization, Hainan Branch of the Institute of Medicinal Plant Development, Chinese Academy of Medical Sciences and Peking Union Medical College, Haikou 570311, China

**Keywords:** *Aquilaria sinensis*, agarwood, heat shock protein 70 family, genome-wide analysis

## Abstract

The heat shock protein 70 (*HSP70*) gene family perform a fundamental role in protecting plants against biotic and abiotic stresses. *Aquilaria sinensis* is a classic stress-induced medicinal plant, producing a valuable dark resin in a wood matrix, known as agarwood, in response to environmental stresses. The *HSP70* gene family has been systematic identified in many plants, but there is no comprehensive analysis at the genomic level in *A. sinensis*. In this study, 15 putative *HSP70* genes were identified in *A. sinensis* through genome-wide bioinformatics analysis. Based on their phylogenetic relationships, the 15 *AsHSP70* were grouped into six sub-families that with the conserved motifs and gene structures, and the genes were mapped onto six separate linkage groups. A qRT-PCR analysis showed that the relative expression levels of all the *AsHSP70* genes were up-regulated by heat stress. Subcellular localization of all HSP70s was predicted, and three were verified by transiently expressed in *Arabidopsis* protoplasts. Based on the expression profiles in different tissues and different layers treated with Agar-Wit, we predict *AsHSP70* genes are involved in different stages of agarwood formation. The systematic identification and expression analysis of *HSP70s* gene family imply some of them may play important roles in the formation of agarwood. Our findings not only provide a foundation for further study their biological function in the later research in *A. sinensis*, but also provides a reference for the analysis of HSPs in other species.

## 1. Introduction

Plants are exposed to dynamic and complex environmental stimuli during their growth, including not only biotic stresses, such as herbivore and pathogen attacks, but also abiotic stresses, such as extreme temperatures, salinity, chilling and drought, causing cell injures and producing secondary stresses. Over the course of long-term evolution, plants have evolved several mechanisms, morphological and physiological adaptations to protect themselves [1,2]. Heat-shock proteins (HSP) are stressed-related proteins and could be induced by a broad spectrum of environmental stresses, such as heat [3], drought [4], salinity [5] and heavy metals [6,7,8]. Since they were first discovered in *Drosophila* in the 1960s, HSPs have been identified in almost all organisms [9]. According to their molecular weight, HSP superfamily proteins are classified into the following five families: HSP60, chaperonin family; HSP70, 70-kDa-heat shock protein; HSP90 family; HSP100 family; sHSP, small heat shock protein [2]. Among these families, HSP70s have drawn the greatest attention as they have housekeeping functions in protein folding and protein quality control by proofreading protein structure and repairing misfolded conformers, preventing protein aggregation or misfolded conformers [10]. Structurally, HSP70s are comprised of the following three major domains: a conserved 44-kD N-terminal ATPase domain (NBD), an 18-kD substrate binding domain (SBD) and a 10-kD variable C-terminal domain. HSP70 are considered to be the most highly conserved HSPs evolutionarily [2,11,12,13].

Diverse biological functions of *HSP70* gene family have been well characterized in several plant systems. Numerous studies have reported HSP70s play vital roles in stress response. For example, it is demonstrated that high expression of *HSP70s* led to tolerance enhancement in several plants [14,15]. In *Nicotiana tabacum*, mutant plants with overexpression of *NtHSP70-1* gene showed high tolerance to drought stress [16]. HSP70s also function with drought and salt tolerance in *Saccharum* spp. *Hybrid* [17], *Dunaliella salina* [18] and *Sorghum bicolor* [19]. Moreover, *HSP70* developmental expression appears to be complex during the growth of vegetative and reproductive stage. Transgenic *cpHSC70-1* in *Arabidopsis* exhibited malformed leaves, variegated cotyledons, impaired root growth, and growth retardation, after heat stress treatment of germinating seeds [20]. In *Arabidopsis*, the cpHSP70s are also essential for maintaining chloroplast development. The *cpHsc70-1/cpHsc70-2* double-mutants show a white and stunted phenotype [21]. *HSP70s* have been well identified in many plant species. So far, 18 copies of *HSP70s* have been reported in *Arabidopsis* [22]. Additionally, 16 members in *Chenopodium quinoa* [23], 18 members in *Pennisetum glaucum* [24] and 61 putative *HSP70* genes in *Glycine max* L. [25] are also reported. 

*Aquilaria sinensis* is a typical stress-induced medicinal plant responding to stress with the formation of dark resinous wood called agarwood, which is widely used in traditional medicine as a digestive, sedative, and antiemetic, as well as being used as a precious scent and perfume across Asia, the Middle East and Europe [26,27,28]. Agarwoods are generally formed only in naturally or artificially injured trees [29,30], and high-quality agarwood is more valuable than gold on a unit weight basis, in the international market. According to the *China Pharmacopoeia*, *A. sinensis* is the only certified source for agarwood [31]. Agarwood formation and accumulation in natural conditions are infrequent and take over decades. Though trees are widespread in South China and Southeast Asia, millions of *Aquilaria* trees are unable to produce agarwood, which makes it a very scarce material. The successful development of a reliable agarwood induction protocol will necessitate a better understanding of the wound-induction molecular mechanism. Our group has previously verified that agarwood formation is the production of defense response in *A. sinensis* [32,33,34,35,36]. HSP70s are a highly conserved protein and perform a fundamental role in protecting plants against abiotic stresses, which means they might play potential function in agarwood formation. However, to date, the *HSP70* gene family members in *A. sinensis* have not been reported. Thus, it is necessary to analyze the protein family to obtain a better understanding of the stress damage defensive mechanism of *A. sinensis*.

In this study, a total of 15 *HSP70* genes were identified using a bioinformatic method based on the genome of *A. sinensis*. Comprehensive analyses were conducted to reveal the gene chromosomal locations, phylogenetic relationships, gene structures and conserved motifs. Moreover, the expression profiles of the *AsHSP70* family in response to heat stress were revealed by the qRT-PCR, and the subcellular localization were studied and verified by transient expression in *Arabidopsis* protoplasts. In addition, the expression patterns of *AsHSP70* genes were detected in different tissues and Agar-Wit treatment in *A. sinensis*.

## 2. Materials and Methods

### 2.1. Identification of HSP70s in Plant Species

To obtain potential members of the *HSP70* gene family in *A. sinensis*, the amino acid sequences of the HSP70s of *Arabidopsis* were downloaded from the TAIR (http://www.arabidopsis.org/, accessed on 28 March 2021), rice HSP70 protein sequences were obtained from the TIGR Database, and the sequences of *Marchantia polymorpha, Selaginella moellendorffii*, *Amborella trichopoda*, *G. max*, *Picea abies*, *Hordeum vulgare* were downloaded from NCBI databases. The eight plant species HSP70 protein sequences were used as queries in basic local alignment search tool (BLATP) search of the *A. sinensis* genome database with the criteria of an e-value < 10^−5^. The results were scanned with the identity ≥50 and align ratio ≥ 50%. A total of 15 *A. sinensis HSP70* gene sequences were obtained after manually filtering out repeated sequences, and sequences without HSP70 domains by HMMSCAN.

### 2.2. Phylogenetic Analyses

Based on the HSP70 protein sequences, ClustalW was used to simultaneously align, the phylogenetic tree was constructed using MEGA7.0 with the Neighbor-Joining (NJ) method, and a bootstrap analysis was conducted using 1000 replicates at each node.

### 2.3. Gene Structure Analysis, Conserved Motif Analysis and Chromosomal Location Analysis

To investigate the AsHSP70 genes diversity and structure, exon and intron structures of AsHSP70 genes were illustrated using Gene Structure Display Server (GSDS, http://gsds.cbi.pku.edu.cn, accessed on 15 June 2021) [37]. The information of AsHSP70 proteins conserved motifs were identified using the MEME program (version 4.11.2, http://alternate.memesuite.org/tools/meme, accessed on 18 July 2021). The chromosomal locations of the AsHSP70 genes were determined using the Populus genome browser (http://www.phytozome.net/poplar, accessed on 16 June 2021). WoLF PSORT software was used to predict the subcellular localization of AsHSP70 proteins based on the chromosomal position information provided in the genomic annotation file [38]. The molecular weight (Da) and isoelectric point (pI) of each gene were obtained using compute pI/Mw tool from ExPASy (http://www.expasy.org/tools/, accessed on 20 July 2021) [39]. The instability index, aliphatic index and Grand average of hydropathicity of the identified proteins were calculated by using ProtParam Tool (http://web.expasy.org/protparam, accessed on 20 July 2021).

### 2.4. Plant Materials and Stress Treatments

*A. sinensis* calli were used as material in this study. All the lines were cultured in dark at 25 °C. For the heat stress, *A. sinensis* calli were removed to a high temperature chamber (42 °C) for 0 h, 6 h, 12 h, 24 h, 36 h, 48 h and 72 h and the calli without any treatment were used as the control. All materials were frozen in liquid nitrogen and stored at −80 °C for further qRT-PCR analysis. The materials for transcription sequence are stems from seven-year-old *A. sinensis*; the trees are grown in wild field in Hainan and were treated using Whole-tree agarwood-inducing technique (Agar-Wit) [40]. Samples were collected at different times. The healthy wood was used as control. 

### 2.5. Subcellular Localization in Arabidopsis Protoplasts

The CDS sequences of AsHSP70-4, AsHSP70-5, AsHSP70-7 and AsHSP70-14 were PCR-amplified with the gene-specific primers, the amplification products were digested with BamHI and Encore, then cloned into the pBWA(V)HS-GFP plasmid, resulting in recombinant expression vectors of AsHSP70s with the CDS of enhanced green fluorescent protein (eGFP) protein. The fusion plasmids were transformed into Arabidopsis protoplasts as Kim has been reported previously [41]. The pBWA(V)HS-GFP empty vector served as the positive control. Images were acquired at 48 h using a Leica DMLe camera (Leica, Wetzlar, Germany).

### 2.6. RNA Isolation and Real-Time qRT-PCR

The total RNA was extracted from treated calli using a Total RNA Rapid Extraction kit RN38-EASYspin Plus (Aidlab, Gdańsk, Poland). An amount of 1 μg of total RNA was reverse-transcribed to cDNA using the PrimeScript™ RT Reagent Kit (Takara, Dalian, China) according to the manufacturer’s protocol. The PCR amplifications were performed using SYBR^®^ Premix Ex Taq™ II (Takara, Dalian, China) on Light Cycler^®^ 480II (Roche Diagnostics, Indianapolis, IN, USA). Each reaction contained 5 μL SYBR^®^ Premix Ex Taq II, 3 μL ddH_2_O, 1 μL cDNA template, and 0.5 μL gene-specific primer in a final volume of 10 μL. The PCR cycling conditions were as follows: 95 °C for 30 s, followed by 40 cycles of 95 °C for 3 s and 60 °C for 30 s. Three independent biological replicates were performed, and the relative expression level for each of AsHSP70s was calculated using the 2^−ΔΔCT^ method [42].

## 3. Results

### 3.1. Identification of the HSP70 Gene Family in A. sinensis

HSP70 sequences of eight plant species were used as queries in BLASTP against protein database in *A. sinensis* with a maximum E-value of 1 × 10^−5^. A total of 15 putative HSP70 gene sequences of *A. sinensis* were finally obtained and named based on the order of their location on the chromosomes. The AsHSP70s encoded proteins varied from 489 to 893 amino acids (aa) in length. Among these proteins, AsHSP70-7 protein sequence was the shortest one with 489 amino acids, and AsHSP70-15 encoded the longest protein with 893 amino acids. The predicted molecular weight of AsHSP70 protein range from 5.3 KD (AsHSP70-7, 489 aa) to 10 KD (AsHSP70-15, 893 aa) and the predicted isoelectric points (PI) values were between 5.12 (AsHSP70-12) and 5.83 (AsHSP70-1). Based on the ExPASy analysis, the predicted protein instability indices showed 3 of the 15 AsHSP70 proteins could be considered as unstable proteins, and the other 12 AsHSP70 proteins were predicted to be stable (cutoff < 40). The detailed information of all AsHSP70 genes is shown in Table 1, and the figure after the localization showed the different possibility of subcellular localization for proteins. To classify these 15 AsHSP70s, the neighbor-joining tree was constructed based on the AsHSP70 protein sequences (Figure 1).

An analysis of the chromosomal location showed that fifteen putative AsHSP70 genes were distributed randomly on six chromosomes and the number on each chromosome was independent to chromosome length (Figure 2). The majority of AsHSP70 genes appeared to congregate at the proximate or the distal ends of the chromosomes. Five predicted AsHSP70 genes were present on chr 6, followed by four genes in chr 2. Only one gene was present in chr 0 and chr 4 with the lowest density. Two genes each were found in chr 6, chr 1 and chr 5.

### 3.2. Phylogenetic Relationship of the HSP70 Genes in A. sinensis

To evaluate the phylogenetic relationships among the *AsHSP70* genes in *A. sinensis* and other plant species in depth, a comprehensive phylogenetic tree of HSP70 proteins from eight different plant species was performed by generating a neighbor-joining phylogenetic tree, including 15 *A. sinensis* HSP70s, 9 *M. polymorpha* HSP70s, 25 *S. moellendorffii* HSP70s, 19 *A. trichopoda* HSP70s, 41 *G. max* HSP70s, 13 *A. thaliana* HSP70s, 27 *O. sativa* HSP70s, 20 *P. abies* HSP70s and 27 *H. vulgare* HSP70s (Table 2). According to previous studies, the number of sub-families in *O. sativa*, *G. max*, *A. thaliana* are 6, 8, 5, respectively [22,25,43]. Based on their phylogenetic relationships, 196 HSP70 proteins were separated into the following six major sub-families: I, II, III, IV, V and VI (Figure 3). Among these six groups, except class VI and IV, all the other groups included nine plant species HSP70 members, as follows: class IV was the smallest sub-family, containing 2 *H. vulgare*, 1 *O. sativa* and 1 *M. polymorpha*; class VI contained 7 members, 2 *S. moellendorffii*, 1 from 5 plant species (*P. abies*, *O. sativa*, *A. trichopoda*, *G. max* and *A. sinensis*), and no *H. vulgare*, liverwort and *A. thaliana*. Cluster V was the largest group containing 76 members, 19 members from *G. max*, 6 from *A. trichopoda*, 2 from *A. thaliana*, 6 from *S. moellendorffii*, 11 from *O. sativa*, 12 from *H. vulgare*, 12 from *P. abies*, 2 from *M. polymorpha* and 6 from *A. sinensis*. Class I contained 41 members was the second largest with 3 *A. sinensis*, 5 *A. thaliana*, 5 *S. moellendorffii*, 5 *A. trichopoda*, 12 *G. max*, 4 *O. sativa*, 3 *H. vulgare*, 2 *P. abies* and 2 *M. polymorpha*. Class III was composed of 39 members (5 *G. max*, 3 *A. sinensis*, 3 *A. thaliana*, 4 *A. trichopoda*, 4 *P. abies*, 6 *S. moellendorffii*, 6 *H. vulgare*, 5 *O. sativa* and 3 *M. polymorpha*). The HSP70 numbers in class II were 4 *G. max*, 2 *A. sinensis*, 3 *A. thaliana*, 3 *A. trichopoda*, 1 *P. abies*, 6 *S. moellendorffii*, 4 *H. vulgare*, 5 *O. sativa* and 1 *M. polymorpha*.

### 3.3. Conserved Motif and Gene Structure Analyses of HSP70 Genes in A. sinensis

To better understand the evolution conservation of AsHSP70 family, the number and distribution of conserved motifs were identified using the MEME program. A total of 10 consensus motifs were detected based on domain compositions of HSP70-conserved amino acid compositions of identified motifs, as shown in Table 3. The results showed that most of the closely related members share similar motif patterns, suggesting possible functional similarity among these AsHSP70 proteins (Figure 4A). For all the 15 AsHSP70 proteins, four motifs (motif 1, motif 3, motif 7, motif 9) existed in all AsHSP70 proteins and motif 2 existed in almost all of them except AsHSP70-8. Except for the lack of motif 10 in AsHSP70-9, proteins from sub-family II, V and III contained all 10 conserved motifs. Over half of AsHSP70 were found to contain more than 19 motifs, while the proteins from sub-family I and VI, including AsHSP70-8, AsHSP70-2, AsHSP70-15 and AsHSP70-5, contained less than 10 motifs. Protein motifs from sub-family V were the most conserved, except AsHSP70-9, due to the fluctuant number and position of motifs. The type, number and position of conserved motifs are revealed be similar in the proteins from the same sub-family.

The genes’ exon–intron structure can reflect the evolution of gene families [44]. To better understand the functional diversification of the AsHSP70 proteins during their evolution, we further investigated the gene structure of 15 *AsHSP70* genes. The results showed that the number of introns within ORFs varied among *AsHSP70* genes, whereas the difference between genes of the same sub-family was small. Genes that belonged in the same cluster showed similar patterns, suggesting these AsHSP70 proteins contained similar function. (Figure 4B). The number of introns ranged from 0 to 14, and more than half of *AsHSP70* genes contain five to nine introns. All members in sub-family V and VI contained one intron, with the exception that no intron was found in two *AsHSP70* genes (*AsHSP70-9* and *AsHSP70-14*). Genes from sub-family I (*AsHSP70-8*, *AsHSP70-12*, *AsHSP70-15*) contained higher numbers of intron than others. *AsHSP70-15* was found to possess the largest number (14) of introns. As for sub-family II and III, the number of introns ranged from five to seven.

### 3.4. Expression Profiles of AsHSP70 Genes in Response to Heat Stress

Heat shock proteins are a kind of protein that are expressed in large quantities after organisms are stimulated by stress, such as high temperature [45], and, among them, HSP70s play vital roles in response to heat treatment. In addition, the existing studies have shown that heat treatment is one of the traditional artificial methods to trigger agarwood formation [46], and our group have reported that heat shock can induce jasmonic acid production and the accumulation of agarwood sesquiterpene in *A. sinensis* cell suspension cultures [47]. Therefore, we first detected the response of AsHSP70s to heat shock treatment. The RNA was isolated from the heat stress-treated calli from different time points (0 h, 6 h, 12 h, 24 h, 36 h, 48 h, 72 h) and the relative expression profiles of 15 AsHSP70 genes were examined by qRT-PCR. As shown in Figure 5, the relative expression level of all 15 AsHSP70 genes were up-regulated by heat stress. The transcriptional levels of eight genes (AsHSP70-2, AsHSP70-3, AsHSP70-9, AsHSP70-10, AsHSP70-11, AsHSP70-12, AsHSP70-13 and AsHSP70-15) share similar expression patterns, they all increased in various degrees and reached the highest levels at 36 h or 48 h after heat stress treatment. The other six genes (AsHSP70-1, AsHSP70-4, AsHSP70-5, AsHSP70-6, AsHSP70-7 and AsHSP70-14) exhibited enhanced expression immediately after heat stress, with slowly or quickly decreased in the later period. Notably, among them, the expression level of three genes (AsHSP70-4, AsHSP70-5 and AsHSP70-7) was dramatically up-regulated (more than 1000-fold) compared with the control in response to heat stress. In contrast, AsHSP70-8 exhibited special expression patterns, its expression did not change and remained stable until 48 h after heat stress, with the following treatment, the expression level of AsHSP70-8 increased rapidly and peaked at 72 h.

### 3.5. Subcellular Localization of AsHSP70s

The protein subcellular localization prediction software WoLF PSORT was used to predict the protein localization of 15 candidate AsHSP70 in *A. sinensis* (Table 1). AsHSP70-10 and AsHSP70-15 were predicted to be localized in the ER. AsHSP70-1 and AsHSP70-3 were predicted to be localized in the mitochondria or in the chloroplast with high reliability, while AsHSP70-4 and AsHSP70-8 were likely localized in vacuole and nucleus, respectively. AsHSP70-6 was predicated to be localized in the chloroplast. For the other eight AsHSP70 proteins (AsHSP70-2, AsHSP70-5, AsHSP70-7, AsHSP70-9, AsHSP70-11, AsHSP70-12, AsHSP70-13 and AsHSP70-14), the cytosol is predicted to be their most likely location. To further confirm their predicted localizations, three genes (AsHSP70-4, AsHSP70-5 and AsHSP70-14) which are up-regulated strongly by heat treatment, were selected for subcellular localization by pBWA(V)HS-GFP translational fusion construct. The recombinant fusion was transiently expressed in *Arabidopsis* leaf protoplast. pBWA(V)HS GFP vector was transformed as the control and was detected in the nucleus and cytoplasm (Figure 6). As shown in Figure 6, the green fluorescence signals from AsHSP70-5 and AsHSP70-14 fusion proteins were both detected in the nuclei and cytoplasm. The prediction results of AsHSP70-5 were consistent with the experimental results, while AsHSP70-14 was not predicated to be localized in the nuclei. AsHSP70-4 fusion protein was localized in the cytoplasm, which was different from the predicted results.

### 3.6. Expression Profiles of AsHSP70 in Different Tissues and Different Layers during the Whole-Tree-Inducing Agarwood Formation Process in A. sinensis

To investigate the temporal and spatial expression profile of the *AsHSP70* genes in different tissues, expression levels of all 15 putative *AsHSP70s* were analyzed based on transcriptome data in 7 tissues, including agarwood, branch, stem, root, old leaves, tender leaves, bud and flower (Figure 7). The results revealed that all *AsHSP70s* genes exhibited broad expression patterns across various tissues, with the exception of the two *AsHSP70s* genes, *AsHSP70-9* and *AsHSP70-7*. *AsHSP70-9* that did not express in agarwood and *AsHSP70-7* that did not express in agarwood and old leaves. The other 13 *AsHSP70s* genes were expressed in all 7 tissues. These findings indicate that *AsHSP70s* genes had diverse function and participated in multiple processes in *A. sinensis*. As a whole, *AsHSP70s* genes showed the highest expression in stem and the lowest expression in tender leaves. As shown in Figure 7A, 15 *AsHSP70s* genes were mainly clustered into four groups (A, B, C, D) according to their expression patterns, which presumes that they have similar function in the same cluster. In cluster A, all genes (*AsHSP70-2*, *AsHSP70-4* and *AsHSP70-7*) exhibited high expression in stem and root, and *AsHSP70-7* also showed high expression in branch. In cluster B and C, seven *AsHSP70s* genes (*AsHSP70-6*, *AsHSP70-8*, *AsHSP70-9*, *AsHSP70-12*, *AsHSP70-13*, *AsHSP70-10*, *AsHSP70-15*) were highly expressed in bud and flower, implying these *AsHSP70s* contained functions in vegetative organs. In cluster D, four *AsHSP70s* genes (*AsHSP70-1*, *AsHSP70-11*, *AsHSP70-3*, *AsHSP70-5*) showed high expression levels in agarwood, as well as *AsHSP70-15*. In contrast, the other three groups were the opposite, suggesting these five *AsHSP70s* genes may be closely related to the formation of agarwood.

To evaluate whether *AsHSP70* genes are involved in the progress of agarwood formation, transcriptome data were obtained from Agar-Wit inducing materials with different treatment time and different layers, the detail sections of agarwood are shown in Figure 8B. As shown in Figure 8A, all *AsHSP70s* were up-regulated in some layers or at certain time, suggesting all of them were involved in agarwood formation and play different functions in the process. In addition, 15 *AsHSP70s* genes were mainly clustered into three groups (A, B, and C) according to their expression patterns. Genes in group B were sensitive to the wound treatment and expressed mainly in decayed layer in early stage (from 2 h to 3 days), suggesting they contribute to cell death in the process of agarwood formation. These results are also consistent with the relative expression treated by heat, whereby genes in group B all exhibited enhanced expression immediately after heat stress. Among them, three genes (*AsHSP70-4*, *AsHSP70-5*, *AsHSP70-7*) expressed by a factor of over 10^3^ after heat treatment. Genes in group C were mainly expressed in the agarwood layer, implying they probably have critical roles in agarwood formation, and their expression induced by heat shared similar patterns as well. Interestingly, during the Agar-Wit treatment, *AsHSP70-8* from group A showed up-regulated expression from 6 h to 3 days, whereas the rest of the time they were downregulated. 

## 4. Discussion

*A. sinensis* is a classic wound-induced medicinal plant and only can form agarwood after being wounded or infected by microbes [35,48]. The molecular mechanism of agarwood formation is still largely unclear. HSP70 is one of the most conserved proteins involved in the process of plant stress tolerance. To verify the functions of *AsHSP70s* in *A. sinensis* in response to stress, we performed a whole-genome analysis of the *AsHSP70* gene family in *A. sinensis* for the first time. Overall, 15 *AsHSP70* genes were identified in *A. sinensis*, which is less than other reported plant species (16 in *C. quinoa*, 18 in *Arabidopsis*, 61 in soybean) [22,23,25]. As shown in Figure 2, the majority of *AsHSP70* genes were unevenly distributed on six chromosomes, mainly on both ends of the chromosome, a pattern that is consistent with that reported in *Nicotiana tabacum* and *A. thaliana* [22,43].

AsHSP70 can be divided into six sub-families in *A. sinensis*. Based on the phylogenetic tree, it was found that proteins from the same sub-family contained a similar type, order, and number of motifs, while these conditions were not seen among different sub-families. In the gene structure analysis, the same rule was also revealed, in that the most closely related *AsHSP70* genes shared a more similar gene structure, while the same results were also found in a number of reported plants [49,50]. Gene structures revealed phylogenetic relationships and related to gene function [51]. The number of introns is usually related to the transcription regulation. Genes in group V and VI contained fewer introns, longer exons and more motifs, while genes in group I contained more introns and fewer motifs. This accords with earlier studies in other plants [43,52]. Genes with fewer introns are able to rapidly activate in timely response to various stresses; plants with a lower number of introns contain a stronger ability to adapt to external stimulus [53,54]. Combined with Figure 4 and Figure 5, genes in sub-family V and VI were highly induced after heat treatment, which supports the standpoints.

Heat treatment is used as an artificial method for producing high-quality agarwood and it can induce jasmonic acid production and the accumulation of agarwood sesquiterpene [47]. In this study, the expression profiles of *AsHSP70* genes under heat treatment revealed that *AsHSP70* genes were involved in responding to heat stress. As results showed, all *AsHSP70* genes were up-regulated in heat stress, some *AsHSP70* genes were induced at early stage, while some of them were induced later, indicating that different *AsHSP70* genes played different roles during heat treatment, which was consistent with the results of expression profiles of different layers and time points treated with Agar-Wit.

HSP70 have been discovered in diverse subcellular compartments, such as the nuclei [4], mitochondrion [15] and endoplasmic reticulum [55]. In this study, AsHSP70 members were predicted located in various cellular compartments in *A. sinensis.* Over half of AsHSP70 proteins were located in cytoplasm or nuclei, similar results were also found in soybean in which more than half of *HSP70* genes were localized in cytosol [25], according with the reports that cytosolic HSP proteins relocate to nuclei to protect cells from heat-induced nuclear damage [56]. In addition, with the analysis of subcellular localization of three AsHSP70 proteins (AsHSP70-5, AsHSP70-4 and AsHSP70-14), all located in the cytosol, two of these proteins (AsHSP70-5, AsHSP70-14) were also located in the nuclei. These results suggest HSP70s are preferred in specific cellular compartments, and AsHSP70s is involved in a defensive reaction with multiple organelles.

As shown in Figure 7, *AsHSP70* genes were expressed in different tissues and organs in *A. sinensis*, indicating that they might participate in various biological processes of plants. Five genes (*AsHSP70-1/3/5/11/15*) were highly expressed in agarwood layers, implying they were promoters of agarwood formation. Combined with the expression patterns of different layers and time points during the agarwood formation (Figure 8A), *AsHSP70-5* and *AsHSP70-11* were expressed mainly in the decayed layer from 2 h to 3 days, then expressed in the agarwood layer at 5 days, while *AsHSP70-1*, *AsHSP70-3* and *AsHSP70-15* mainly expressed in the agarwood layer at 24 h. These results are consistent with their expression patterns induced by heat shock. In addition, we further explore the potential role of AsHSP70 in agarwood formation through the expression profiles in the process of agarwood formation treated with Agar-Wit. As shown in Figure 8A, genes in group B (*AsHSP70-4*, *AsHSP70-5, AsHSP70-6*, *AsHSP70-7*, *AsHSP70-11* and *AsHSP70-14*) expressed specifically in the decayed layer at early stage (from 2 h to 24 h), implying they may play a regulatory role in cell death in the process of agarwood formation. Heat shock proteins help plants recover from stress through repairing damaged proteins or degrading them, thus restoring protein homeostasis and promoting cell survival. The results of qRT-PCR also showed that they were all induced immediately. Genes in group C (*AsHSP70-1*, *AsHSP70-2*, *AsHSP70-3*, *AsHSP70-10*, *AsHSP70-12* and *AsHSP70-15*) expressed highly in the agarwood layer at 24 h, and all of them showed similar expression patterns after heat treatment, suggesting they might play a positive regulator in producing agarwood. These results suggest that the regulatory mechanism of agarwood formation is complicated, and that different kinds of AsHSP70 proteins play various function in different stages of the agarwood formation process.

## 5. Conclusions

In this study, we performed a comprehensive analysis of the *HSP70* genes in *A. sinensis* covering phylogeny, chromosomal location, conserved motifs, gene structures, expression profiles and subcellular localization. In total, 15 *AsHSP70* genes were identified and classed into six groups. The motifs and gene structures within the same subfamilies shared a similar pattern. The qRT-PCR results showed that the expression pattern of 15 *AsHSP70* genes were all affected by heat stresses. Combined with the expression profiles of different tissues, and expression patterns in the different layers after the Agar-Wit treatment, different *AsHSP70* genes are involved in different stages of formatting agarwood and contribute to various biological functions. This study lays a solid foundation for further functional analyses of *AsHSP70* genes in *A. sinensis*.

## Figures and Tables

**Figure 1 genes-13-00008-f001:**
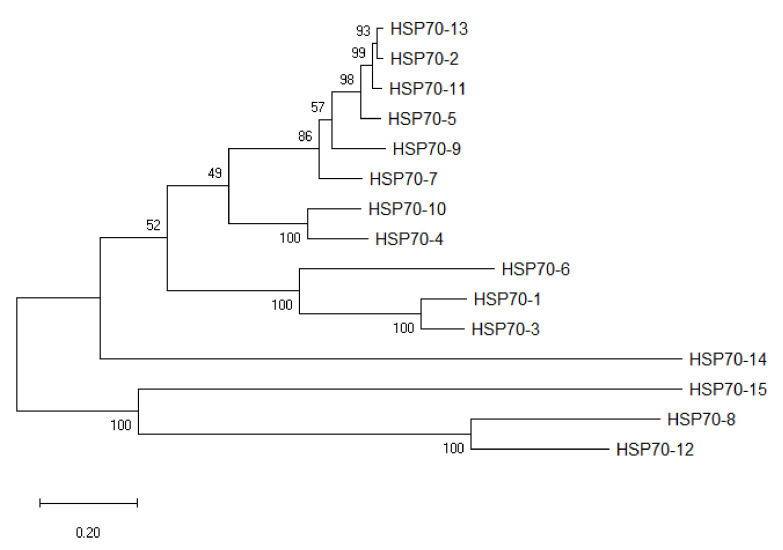
Phylogenetic tree of AsHSP70s in *A. sinensis*. Fifteen AsHSP70 members were identified from the genome in *A. sinensis*. Unrooted phylogenetic tree was constructed based on multiply-aligned sequences of the 15 AsHSP70 proteins with 1000 bootstrap replications.

**Figure 2 genes-13-00008-f002:**
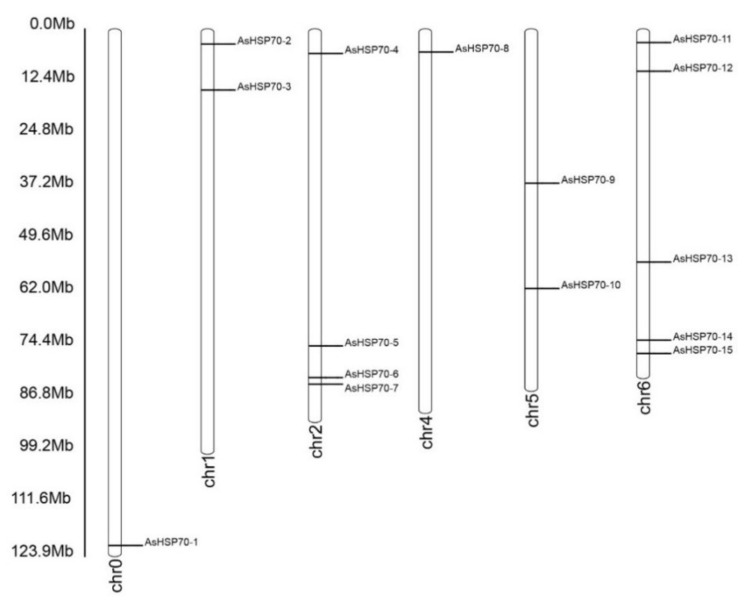
Chromosomal location of AsHSP70 in *A. sinensis*. The chromosome number is indicated at the bottom of each chromosome. The bar located on the left side indicates the chromosome sizes in mega bases (Mb).

**Figure 3 genes-13-00008-f003:**
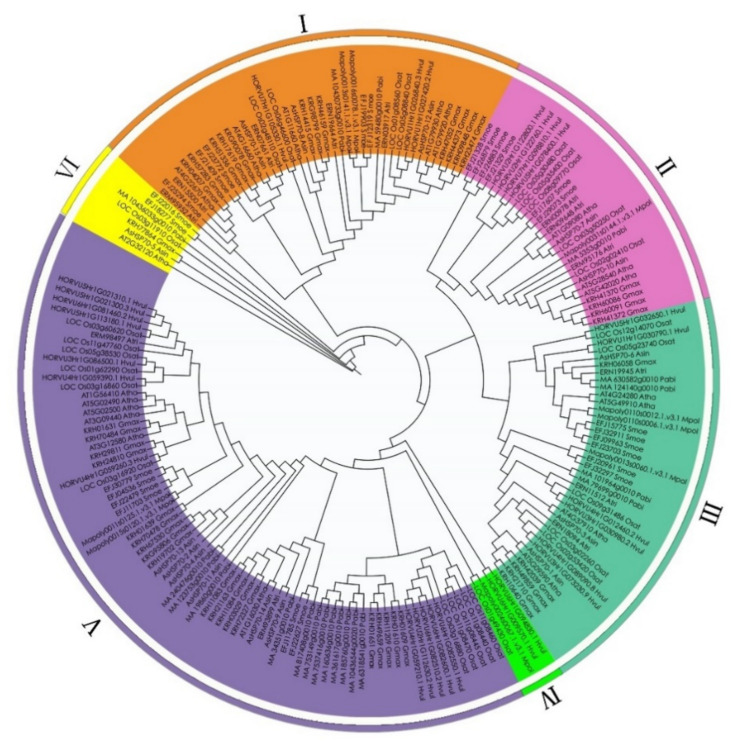
Phylogenetic relationship of HSP70 in *A. sin* (*Aquilaria sinensis*), *A. tha* (*Arabidopsis thaliana*), *A. tri* (*Amborella trichopoda*), *G. max* (*Glycine max*), *H. vul* (*Hordeum vulgare*), *M. pol* (*Marchantia polymorpha*), *O. sta* (*Oryza sativa*), *P. abies* (*Picea abies*), *S. moe* (*Selaginella moellendorffii*). Based on the total identified 196 HSP70 homologs in 9 plant species, an unrooted phylogenetic tree was calculated with the maximum likelihood method, using JTT modeling with γ-distributed rates and 1000 bootstrap replications.

**Figure 4 genes-13-00008-f004:**
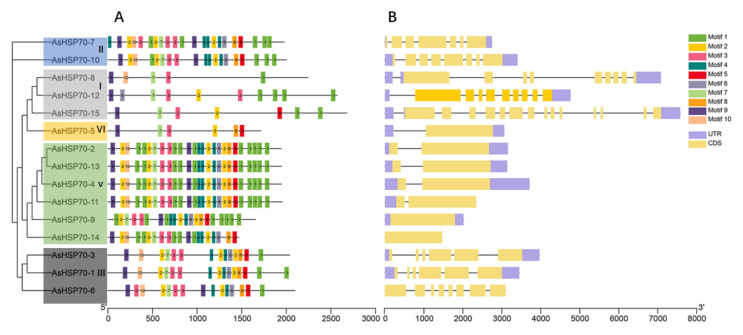
Distribution of conserved motifs and gene structure of the *AsHSP70*. (**A**) Conserved motifs analysis of AsHSP70 proteins using MEME tools. Conserved motifs are shown in different colored boxes. (**B**) Exon–intron organization of *AsHSP70* genes. Yellow boxes represent exons and black lines represent introns. The untranslated regions (UTRs) are indicated by purple boxes.

**Figure 5 genes-13-00008-f005:**
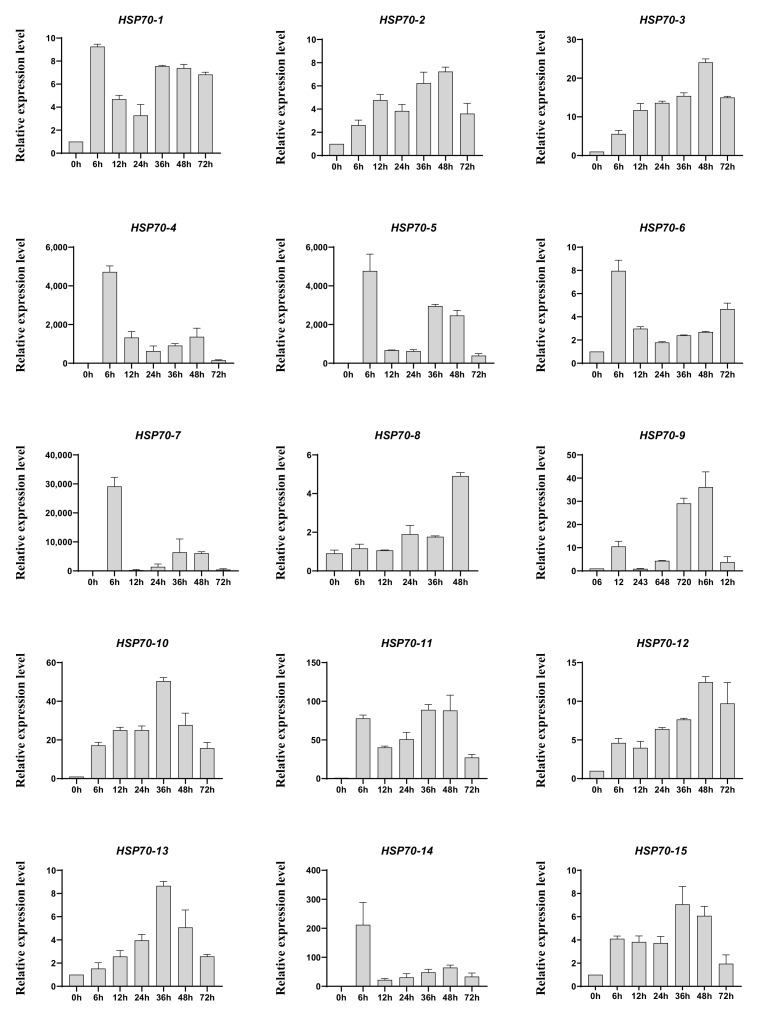
Expression profiles of *AsHSP70s* in heat treated calli. *A. sinensis* calli were transferred from 25 °C to 42 °C and sampled at appointed times (0 h, 6 h, 12 h, 24 h, 36 h, 48 h, 72 h). Expression levels of all detected genes were assayed using real-time PCR analysis and *AsGADPH* as the internal control. Each value is the mean ± SE of 3 independent biological replicates.

**Figure 6 genes-13-00008-f006:**
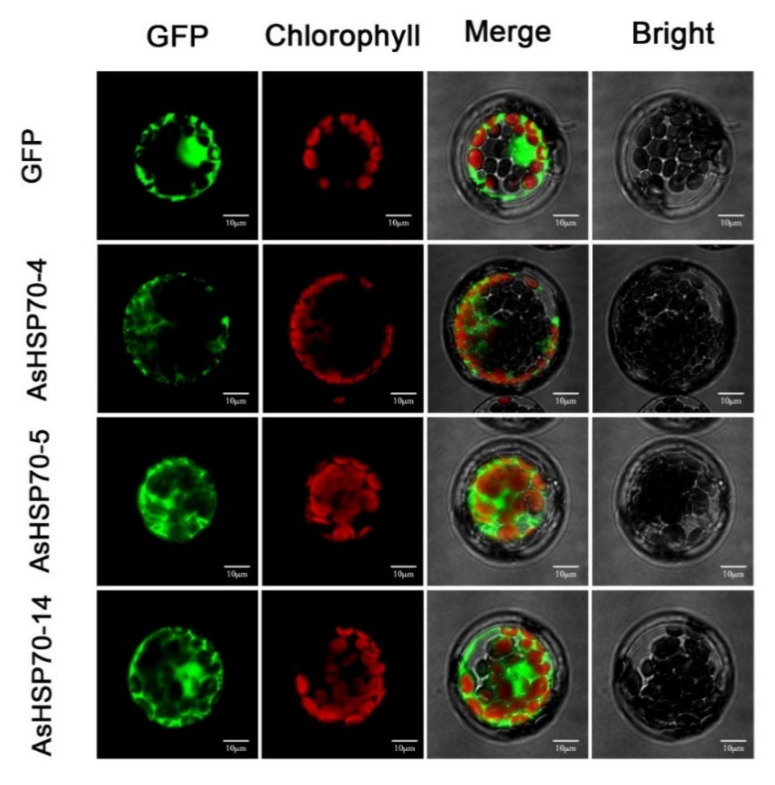
Subcellular localization of AsHSP70 proteins in *Arabidopsis* protoplast. The AsHSP70-4-pBWA(V)HS-GFP, HSP70-5-pBWA(V)HS-GFP, HSP70-14-pBWA(V)HS-GFP and pBWA(V)HS-GFP vectors were transiently expressed in *Arabidopsis* protoplast. Confocal images were acquired at 48 h. Scale bar = 10 μm.

**Figure 7 genes-13-00008-f007:**
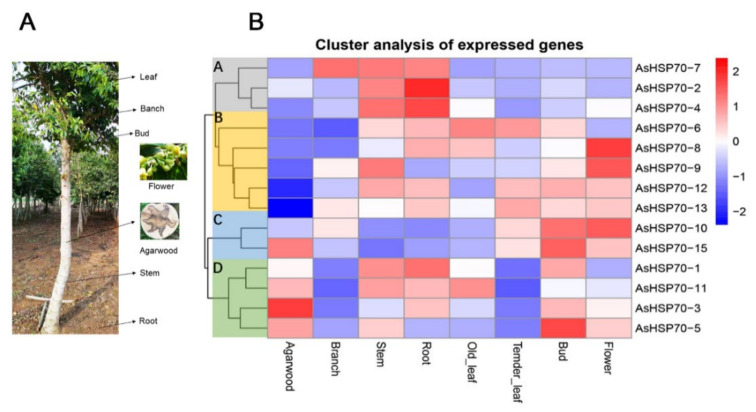
Heat map of the *AsHSP70* genes expression profile in different tissues and different tissues of *A. sinensis* trees. (**A**). The different tissues of *A. sinensis* trees. The agarwood layer was obtained after trees was wounded by external stimulus. (**B**). All gene expression levels were transformed to scores ranging from −2 to 2 and were colored blue, white, or red to represent low, moderate, or high expression levels, respectively.

**Figure 8 genes-13-00008-f008:**
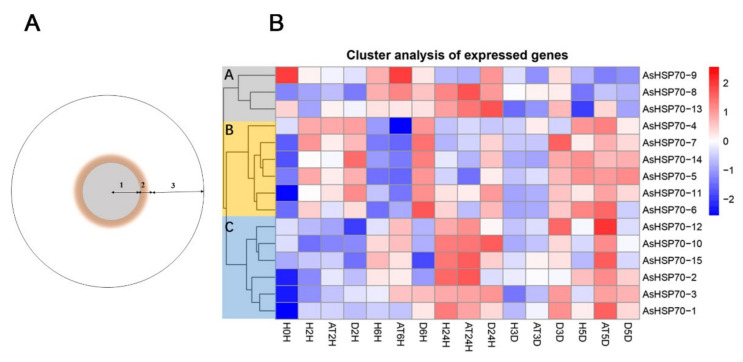
Heat map of the *AsHSP70* genes expression profile in different times under Agar- Wit treatment and different layers of agarwood in cross section of *A. sinensis* tree. (**A**) 1. the decayed layer after Agar-Wit treatment (D); 2. the agarwood and transition layer (AT); 3. the healthy layer (H). (**B**) All gene expression levels were transformed to scores ranging from −2 to 2 and were colored blue, white or red to represent low, moderate, or high expression levels, respectively.

**Table 1 genes-13-00008-t001:** Summary information of predicted physiological and biochemical properties of the AsHSP70 proteins based on the ExPASy analysis.

Name	Amino Acids	MW (kDa)	Exon Number	pI	GRVAY	Instability Index	Aliphatic Index	Subcellular Localization
AsHSP70-1	676	72.71	6	5.83	−0.292	38.21	89.2	mito: 8; chlo: 6
AsHSP70-2	647	70.88	2	5.16	−0.408	35.04	82.63	cyto: 8; cysk: 5, chlo: 1
AsHSP70-3	679	72.77	6	5.81	−0.292	38.97	88.25	mito: 11; chlo: 3
AsHSP70-4	658	73.25	2	5.22	−0.487	31.09	88.34	vacu: 4, chlo: 3, mito: 3, golg: 2, nucl: 1, extr: 1
AsHSP70-5	650	71.10	1	5.21	−0.415	31.54	81.65	cyto: 10; cysk: 2; chlo: 1; nucl: 1
AsHSP70-6	699	74.66	8	5.3	−0.299	28.82	87.58	chlo: 14
AsHSP70-7	489	53.74	7	6.38	−0.269	37.64	86.36	cyto: 10; cysk: 3; chlo: 1
AsHSP70-8	747	83.43	9	5.19	−0.287	40.53	89.02	nucl: 7; vacu: 1; cysk: 2; chlo: 2; cyto: 2
AsHSP70-9	551	61.37	1	5.46	−0.422	39.27	88.82	cyto: 9; cysk: 3, chlo: 2
AsHSP70-10	667	73.54	8	5.2	−0.449	27.24	88.58	E.R.: 14
AsHSP70-11	648	70.94	2	5.16	−0.422	33.08	82.19	cyto: 8, cysk: 5, chlo:
AsHSP70-12	856	94.75	9	5.12	−0.472	40.68	77.94	cyto: 8; chlo: 3; nucl: 2; cysk: 1
AsHSP70-13	648	70.98	2	5.17	−0.407	34.1	82.35	cyto: 9, cysk: 3, chlo: 2
AsHSP70-14	571	61.90	1	5.4	0.075	35.79	103.26	cyto: 9; chlo: 4; vacu: 1
AsHSP70-15	893	100.00	14	5.26	−0.45	42.41	86.7	E.R. 14

Abbreviations: MW: molecular weight; PI: isoelectric point; GRVAY: grand average of hydropathicity, chlo: chloroplast, cyto: cytoplasm, E.R.: endoplasmic reticulum, golg: golgi apparatus, mito: mitochondria, nucl: nucleus, vacu: vacuole, cysk: cytoskeleton, extr: extracellular.

**Table 2 genes-13-00008-t002:** HSP70 family members in selected plant species.

Plant Species	Phylogenetic Class						Number
	I	II	III	IV	V	VI	
*Aquilaria sinensis*	3	2	3		6	1	15
*Arabidopsis thaliana*	5	3	3		2		13
*Amborella trichopoda*	5	3	4		6	1	19
*Glycine max*	12	4	5		19	1	41
*Hordeum vulgare*	3	4	6	2	12		27
*Marchantia polymorpha*	2	1	3	1	2		9
*Oryza sativa*	4	5	5	1	11	1	27
*Picea abies*	2	1	4		12	1	20
*Selaginella moellendorffii*	5	6	6		6	2	25

**Table 3 genes-13-00008-t003:** HSP70 motif sequences identified in *A. sinensis* by MEME tools.

Motifs	Width (aa)	Best Possible Match
Motif 1	50	SRAGRDGRWBVAGVWKGCKGAGRAGKWYNWGDSNRAGGACAAGAAGRASA
Motif 2	50	RTHRWCMCYGWKSMDGCYDWYKYYWATGRYGCYSMRRDYCARGCHRCHAW
Motif 3	50	RAVGKBRAGRMVAMYRYYGKDGWYWYYSAYCTBGGTGGTGRDRMYTTTGA
Motif 4	38	RAVRARGTCCADGANNTKSTKCTKBTSGDTGKKACTHC
Motif 5	50	GGCATYCCWCCWGCNCCAAGRGGWGTTCCKCARATHRAWGTBWSMTTYGA
Motif 6	50	GGHMTKGARACTGYTGGTGGWGTNATGACHRWRTTGATTCCHAGRAACAC
Motif 7	50	ATTGCDGGSYTDAAWGTNVTKAGDATVATYAATGAGCCMACWGCWGCTGC
Motif 8	50	ATYMARGTMTWYGAAGGTGARAGAGCRAKRRCMARRGAYAAYAAMTTKCT
Motif 9	50	GGDATTGAYTTCKRYWCSACNWWYWCYHGTGYYRGHKTYKAGGAGCDSRA
Motif 10	50	AACCCMRMRAACACMRTYTTCGAKGYCAAGMGKYTBATYGGHAGDARATT

## Data Availability

Not applicable.

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
