# Peer review of "Genome-Wide Identification and Characterization of HSP70 Gene Family in Aquilaria sinensis (Lour.) Gilg"

_genes, 2021, doi:10.3390/genes13010008_

Round 1

Reviewer 1 Report

The manuscript presented is related to identification and characterization of heat shock proteins in Aquilaria sinensis. Authors carefully predict and verify subcellular localization of all identified genes. The findings provide a lot of new data on systematic and expression patterns of the gene family, but it would be liked to hear some ideas on how the information presented will help to obtained agarwood.

Several reductions in the Table 1 are not explained in the text. Among them “cysk and extr”. And what does the figure after the localization means?

Author Response

Response to Reviewer 1 Comments

Point 1: The manuscript presented is related to identification and characterization of heat shock proteins in Aquilaria sinensis. Authors carefully predict and verify subcellular localization of all identified genes. The findings provide a lot of new data on systematic and expression patterns of the gene family, but it would be liked to hear some ideas on how the information presented will help to obtained agarwood.

Response 1: The expression of AsHSP70 genes in different tissues, suggesting some  AsHSP70s genes that expressed highly in agarwood layer may be closely related to the formation of agarwood. To evaluate whether AsHSP70 genes are involved in the progress of agarwood formation, transcriptome data was obtained from Agar-Wit inducing materials with different treatment-time and different layers, all AsHSP70s were up-regulated in some layers or at certain time, suggesting all of them were involved in agarwood formation and play different functions in the process. Furthermore, heat treatment is used as artificial methods for producing high-quality agarwood, here, we confirmed that all AsHSP70 genes were up-regulated in heat stress. All results above suggest AsHSP70s genes might the positive regulator in producing agarwood, while the detail molecular mechanism of regulating agarwood formation need further explore, and our group will pay attention to how to use AsHSP70s to improve the efficiency of agarwood formation.

Point 2: Several reductions in the Table 1 are not explained in the text. Among them “cysk and extr”.

Response 2: I have explained the missing reductions in the Table 1 in manuscript.

Point 3: What does the figure after the localization means?

Response 3: The figure after the localization is the score that calculated by WoLF PSORT based on both known sorting signal motifs and some correlative sequence features. Based on the score, we could speculate the possible subcellular localization of protein.

Reviewer 2 Report

The subject targeted by the authors and methods used to obtain experimental results were deemed to be of high merit. The paper is generally well-conceived and written but many grammatical errors and questionable terms were found that should be remediated. The reviewer conducted more exhaustive scrutiny of the Abstract and Introduction sections to exemplify these issues. Scrutiny was less rigorous in the other sections, but the authors should carefully consider the comments pertaining to the Abstract and Introduction when rewriting the other sections. A more detailed list of these issues is as follows:

Abstract

Line 17) “…forming dark valuable resin wood named agarwood in response to external stimulus.” Suggest change to “…producing a valuable dark resin in a wood matrix, known as agarwood, in response to environmental stresses.”

Lines 17-18) “HSP70 family has been systematic identified…” change to “The HSP70 gene family has been identified…”

Lines 20-21) “…the 15 AsHSP70 were classified into…” change to “…the 15 AsHSP70 genes were grouped into…”

Line 22) “…they were mapped on 6 chromosomes.” Change to “…the genes were mapped onto six separate linkage groups.”

Introduction

Line 35) “Plants are exposed to complicated environment during their growth, not only biotic…” change to “Plants are exposed to dynamic and complex environmental stimuli during their growth, including not only biotic…”

Line 40) “…induced by kinds of environmental…” change to “…induced by a broad spectrum of environmental…”

Line 41) “Since initially…” change to “Since they were first…”.

Line 42) “…HSP has been…” change to “…HSP have been…”.

Line 51) “…and they are considered to be the most highly conserved HSP…” reviewer suggests this be rendered a stand-alone sentence: “HSP70 are considered to be the most highly conserved HSPs evolutionarily…”

Line 59) “Moreover, HSP70s appear complex developmental…” change to “Moreover, HSP70 developmental expression appears to be complex…”

Line 69) “…which widely used…” change to “…which is widely used in…”

Line 71) “Agarwood are…” change to “Agarwoods are…”

Lines 72-72) “…agarwood is worth than gold…” change to “…agarwood is more valuable than gold on unit weight basis…”

Lines 77-78) “To obtain efficient induction technology, fully understand the wounding-induced molecular mechanism…” suggest change to “The successful development of a reliable agarwood induction protocol will necessitate a better understanding of the wound-induction molecular mechanism…”

Line 84) “…of the defensive mechanism…” change to “…of the stress damage defensive mechanism…”

Results

Lines 95 and 96) “…A. sinensis…” should be italicized “…A. sinensis…”.

Line 108) “…locating…” change to “…location…”.

Line 139 and elsewhere in the paper) “…H. vulgar…” change to “…H. vulgare…”.

Lines 175-176) “Since exon-intron structure can provide an important insight into the evolution of gene families [38].” This sentence is fragmentary and must be rewritten.

Line 180) “…bellowed…” the reviewer is not familiar with the use of this verb in context. Do the authors intend to say “…belonged…”?

Line 197) “…, among them,…” change to “…and, among them,…”.

Line 200) “…JA…” was this acronym defined earlier in the paper? If not, it should be defined here. The MM section appears at the end of the manuscript.

Line 213) “…were…” change to “…was…”.

Line 215) “…maintained stable…” change to “…remained stable…”.

Line 234) “…-ment were selected…” change to “…-ment, were selected…”.

Line 238 and elsewhere in the paper) “…nuclear…” change to “…nuclei…”.

Line 241) “…the predication results.” Change to “…predicted results.”

Line 254 and elsewhere in the paper) “…did not express…” change to “…that did not express…”.

Line 258 and elsewhere) “…amount…” delete.

Lines 282-283) “These results also accordance with…” change to “These results are also consistent with…”.

Line 285) “…expressed over 1000 times…” change to “…expressed by a factor of over 103…”

Line 289) “…, the rest of time they…” change to “…; the rest of the time they…”.

Discussion

Line 300) “…protein…” change to “…proteins…”.

Lines 306-307) “…which consistent with that in Nicotiana tabacum and A. thaliana.” Change to “…a pattern that is consistent with that reported in Nicotiana tabacum and A. thaliana.”.

Line 308) “HSP70 is a multigene family,..” Delete.

Lines 310-311) “…while this situation did not find among different sub-family.” Change to “…while these similarities were not seen among sub-families.”.

Line 315 and elsewhere) “…lesser…” should this be changed to “…fewer…”?

Line 317) “Which is accord with the earlier studies in other plants [37, 46].” Change to “This is accord with the earlier studies in other plants [37, 46].

Line 325) “…shown…” change to “…showed…”.

Line 344) “…positive regulator in…” change to “…promoters of…”.

Line 348) “…mainly expressed…” change to “…expressed mainly in…”.

Line 355) “…recovery…” change to “…recover…”.

Conclusions

Line 427) “…that 15…” change to “…that the expression patterns of 15…”.

Author Response

Response to Reviewer 2 Comments

Point 1: The subject targeted by the authors and methods used to obtain experimental results were deemed to be of high merit. The paper is generally well-conceived and written but many grammatical errors and questionable terms were found that should be remediated. The reviewer conducted more exhaustive scrutiny of the Abstract and Introduction sections to exemplify these issues. Scrutiny was less rigorous in the other sections, but the authors should carefully consider the comments pertaining to the Abstract and Introduction when rewriting the other sections.

Response 1: Thanks a lot for your detailed corrections of grammatical errors and questionable terms. I have corrected these errors and other similar questionable terms according to your suggestions.